# Improving Positioning Accuracy via Map Matching Algorithm for Visual–Inertial Odometer

**DOI:** 10.3390/s20020552

**Published:** 2020-01-19

**Authors:** Juan Meng, Mingrong Ren, Pu Wang, Jitong Zhang, Yuman Mou

**Affiliations:** 1College of Automation, Faculty of Information Technology, Beijing University of Technology, Beijing 100124, China; S201761125@emails.bjut.edu.cn (J.M.); wangpu@bjut.edu.cn (P.W.); a17812102961@163.com (J.Z.); mu3218509460@163.com (Y.M.); 2Engineering Research Center of Digital Community, Ministry of Education, Beijing 100124, China; 3Beijing Key Laboratory of Computational Intelligence and Intelligent Systems, Beijing 100124, China

**Keywords:** visual–inertial odometer, indoor positioning system, conditional random field, map matching

## Abstract

A visual–inertial odometer is used to fuse the image information obtained by a vision sensor with the data measured by an inertial sensor and recover the motion track online in a global frame. However, in an indoor environment, geometric transformation, sparse features, illumination changes, blurring, and noise will occur, which will either cause a reduction in or failure of the positioning accuracy. To solve this problem, a map matching algorithm based on an indoor plane structure map is proposed to improve the positioning accuracy of the system; this algorithm was implemented using a conditional random field model. The output of the attitude information from the visual–inertial odometer was used as the input of the conditional random field model. The feature function between the attitude information and the expected value was established, and the maximum probabilistic value of the attitude was estimated. Finally, the closed-loop feedback correction of the visual–inertial system was carried out with the probabilistic attitude value. A number of experiments were designed to verify the feasibility and reliability of the positioning method proposed in this paper.

## 1. Introduction

Mobile robots are widely used in industrial production, the service industry, and other fields. Mobile robots need to realize autonomous navigation instead of being manipulated by human beings. To achieve this purpose, positioning is one of the key technologies. The global positioning system (GPS) can be used accurately in an outdoor environment; however, in an indoor environment, the global navigation satellite system (GNSS) signal is weak; thus, it cannot complete the positioning function [1]. At present, the commonly used indoor positioning technologies are laser, inertial navigation system, infrared, and wireless local area network (WLAN), but these indoor navigation technologies cannot be widely used because of problems such as allele accuracy and cost. With the rapid development of machine vision and computer technology, the performance of small industrial cameras and the microelectromechanical system (MEMS) inertial devices was continuously improved [2,3,4]. The technology of a visual–inertial odometer (VIO) was gradually realized in engineering applications at the theoretical verification stage.

A monocular visual–inertial navigation algorithm was developed rapidly in recent years. In 2016, Usenko, from the Munich University of Technology, proposed a binocular visual–inertial positioning method based on tight coupling and direct methods. Using the photometric errors and residuals of images, an objective function was established, and a semi-dense map was constructed to achieve a positional accuracy of less than 0.1% of the motion distance [5]. Mur-Artal, author of ORB-SLAM, and others also proposed the addition of an inertial measurement unit (IMU) error model to the cost function of tracking and the local bundle adjustment (BA) module to improve the positioning accuracy in 2016 [6]. The VINS-MONO system proposed by Shaojie’s research group from the Hong Kong University of Science and Technology in 2017 is a VIO system based on sliding window optimization. It only optimizes the information in the sliding window and does not optimize the previous state or measurement value. Its translation error based on loop detection is less than 0.3% of the accuracy of the motion distance [7]. On one hand, all of these algorithms need rich features in the scene. On the other hand, they do not optimize the previous state or measurement value. In an indoor environment, corridors, white walls, and other weak-texture scenarios, as well as the influence of the illumination intensity, make the existing fusion algorithms less robust, sharply decline the precision, or even cause location failure.

The indoor environment has spatial constraints and can be used to eliminate some incorrect positioning results. It is an additional data source to improve the accuracy and reliability of an indoor positioning system. The process of utilizing map information in the positioning process is called map matching. This method can establish a matching model through the corresponding relationship between the attitude sequence obtained by the visual–inertial positioning system and the sequence of the state points in the map; therefore, the positioning accuracy of the system can be improved without adding any hardware [8].

In an indoor map matching algorithm, points (doors), lines (grids), and surfaces (rooms) need to be constrained. In Reference [9], the indoor structure model was constructed by using the indoor basic vector information, and the wireless location results were constrained by fusing the sensor information and the map information. However, the final location results were constrained by the map in this document. The smoothing filtering problem of the location results was not considered in the constraints process. At present, the common methods are based on a hidden Markov model (HMM) and a particle filter (PF) [10,11,12], because of its non-parameterized characteristics. Here, the stochastic variables must satisfy the Gaussian distribution when solving the problem of nonlinear filtering, which provides an effective solution for the analysis of nonlinear dynamic systems. One of the major issues of the PF is the computation time, as a large number of particles are typically required to ensure a good estimation of the continuous probability distribution, particularly when dealing with noisy inertial data and large maps. The MapCraft system proposed by Xiao is the first in which a map matching algorithm based on a conditional random field is tightly coupled with the positioning system [13]. The system obtains the original measurement value directly from the sensor as the input of the algorithm and fuses it with the map information. When this system is used for real-time tracking, the delay caused by the backward phase of the conditional random field will affect the efficiency of the algorithm and reduce the positioning accuracy. Therefore, in this study, we developed a map matching algorithm based on a conditional random field model, which used only the output of the position information from a visual–inertial system as the input of the conditional random field model and constrained it with the map information. The error in the position updating process was reduced by the feedback correction method, and the positioning result of the visual–inertial system was continuously corrected to make it more accurate. Finally, the algorithm was validated experimentally. The experimental results showed that the algorithm yielded good results with respect to improving the positioning error of the visual–inertial system.

## 2. System

Figure 1 shows the system structure diagram. The system consists of four subsystems. The first part is the visual–inertial system, which is the main positioning system to generate the estimated trajectory. In this study, the open-source VINS-MONO system introduced in 2017 was adopted by the Shaojie Research Group. The second part involves map preprocessing, mainly in the form of the original map, which generates the map types required by the map matching algorithm. The indoor map needs to be discretized to evenly spread the discrete points over the reachable area and get the matched state points. The third part is a map matching algorithm based on the conditional random field. The estimated trajectory is fused with the data of the map processing system by the conditional random field, and an accurate matching trajectory is obtained. The matching result is used as feedback to correct the position of the VINS-MONO system at any moment. The fourth part is the output of the final corrected position trajectory.

### VINS-MONO System Overview

VINS-MONO is a real-time VI-SLAM system that integrates inertial sensor data and monocular vision through tight coupling. The system diagram is shown in Figure 2. The system can be operated in a small indoor environment or a wide range of outdoor environments with strong robustness and stability. Firstly, a robust initialization scheme is proposed to provide a relatively accurate initial value for monocular tightly coupled VIO, including the restoration of the visual structure scale, gyroscope offset calibration, and velocity and gravity estimation, which can make the system from the unknown begin to run stably. Secondly, with the combination of the pre-integrated IMU measurements and visual feature observations, a tightly coupled nonlinear optimization method is used to obtain a high-precision visual–inertial odometer. Lastly, the system also performs closed-loop detection and global map optimization. In VIO, there is only cumulative drift on (x,y,z) and the yaw angle; therefore, only these four degrees of freedom are optimized. VINS-MONO has good robustness and accuracy.

## 3. Design of Map Matching Algorithms

### 3.1. Preprocessing of Indoor Maps

Indoor maps come in a variety of formats, such as image file formats, PDF files, or CAD files. These formats cannot be directly used for map matching algorithms, and the map needs to be digitized to create mathematical models. The map in this article is in the label image file format (TIFF). We used the mapinfo software to digitize the electronic version of the indoor map, using only the wall information of the map. The digital information format features the coordinates of the end point of each line segment.

Map matching can be divided into two types: point-to-point matching and trajectory matching. The point-to-point matching method matches the position point with the position of the indoor space according to the floor plan. This method is simple and computationally efficient. Trajectory matching is the matching of the motion trajectory obtained by the initial positioning system with the geometric topology information of corners, corridors, and rooms. This method is highly accurate but computationally intensive. In this study, the map matching of the conditional random field model was adopted, and the point-to-point matching method was more suitable for the visual–inertial positioning system. The method covered the entire indoor area with equally spaced points and stored the coordinates of each state point [9]. At the same time, we removed the state point of the unreachable area. In Figure 3, the red points indicate a state point, while the red lines indicate an indoor structure.

### 3.2. Conditional Random Field Model

The conditional random field model was proposed by Lafferty et al. in 2001 [14]. Conditional random fields (CRFs) are a type of discriminative undirected probabilistic graphical model. They are used to encode known relationships between observations and construct consistent interpretations, and they are often used for the labeling or parsing of sequential data, such as natural language processing or biological sequences, and in computer vision [15,16,17,18,19].

A conditional random field can be viewed as an undirected graphical model or a Markov random field. Ideally, if the description of the sequence of states has conditional independence, the structure of the graph can be arbitrary. However, when building the model, we usually choose the linear-chain undirected graph structure, which is the most commonly used, as shown in Figure 4.

In this study, we defined $o=o_{1},o_{2},\cdots ,o_{n}$ for a given sequence of observations (a sequence of n observation points). $y=y_{1},y_{2},\cdots ,y_{n}$ is the output state point sequence, which is the predicted robot motion trajectory. Then, the output sequence conditional probability can be defined as shown in Equation (1).
(1)$$P(y|o)=\frac{1}{Z(o)}\exp\left(\sum_{j}\lambda_{j}t_{j}(y_{k-1},y_{k},o,k)+\sum_{i}\mu_{i}s_{i}\left(y_{k},o,k\right)\right).$$
(2)$$Z(o)=\sum_{Y}\exp\left(\sum_{j}\lambda_{j}t_{j}(y_{k-1},y_{k},o,k)+\sum_{i}\mu_{i}s_{i}\left(y_{k},o,k\right)\right).$$

Equation (1) denotes that, under the o condition, the joint distribution form of sequences y can be evaluated. Z(o) is a normalizing factor, which can be calculated using Equation (2). $\lambda_{j}$ and $\mu_{i}$ are the feature weights that can be determined by training the model. In this study, we set both weights to one. In general, the value of the feature function is zero or one; that is, the dissatisfaction feature condition is zero; otherwise, it is one. As summarized in the review, the value of the linear-chain condition random field depends mainly on the characteristic coefficient and the feature function.

In Equation (1), both $t_{j}(y_{k-1},y_{k},o,i)$ and $s_{i}(y_{k},o,i)$ are feature functions; further details follow in Section 3.3 and Section 3.4, but the meanings are different. A function corresponds to the corresponding meaning at each position. The same function can be added at different positions to convert the original local features functions into global feature functions. The conditional random field expression can then be rewritten as a vector form, also known as a conditional random field simplification. The previous local feature function can be uniformly written as a global feature function. Assuming that the conditional random field contains $N_{1}$ transfer functions and $N_{2}$ observation functions, then the conditional random field contains $N=N_{1}+N_{2}$ global eigenfunctions.
(3)$$f_{n}(y_{k-1},y_{k},o,i)=\{\begin{array}{c} t_{j}(y_{k-1},y_{k},o,k),\;n=1,2,\ldots ,N_{1}\\ s_{i}(y_{k},o,k),\;n=N_{1}+i;i=1,2,\ldots ,N_{2} \end{array}.$$

Then, the sum of the global feature functions at various locations k can be calculated as follows:(4)$$f_{n}(y,o)=\sum_{k=1}^{k}f_{n}\left(y_{k-1},y_{k},o,k\right),\;n=1,2,\ldots ,N.$$

Similarly, the weight of the local function can be replaced with the global weight as follows:(5)$$w_{n}=\{\begin{array}{cc} \lambda_{j}, & j=1,2,\ldots ,N_{1}\\ \mu_{i}, & i=N_{1}+i,i=1,2,\ldots ,N_{2} \end{array}.$$

Then, Equations (1) and (2) can be expressed as follows:(6)$$P(y|o)=\frac{1}{Z(o)}\exp\sum_{n=1}^{n}w_{n}f_{n}\left(y,o\right),$$
(7)$$Z(o)=\sum_{y}\exp\sum_{n=1}^{n}w_{n}f_{n}\left(y,o\right).$$

We can denote weight $w_{n}$ in the vector form $w={\left(w_{1},w_{2},w_{3},\ldots ,w_{n}\right)}^{T}$; thus, the global feature function can be composed into a vector; then, $F(y,o)={\left(f_{1}(y,o),f_{2}(y,o),\ldots ,f_{n}(y,o)\right)}^{T}$. Therefore, Equations (6) and (7) can be rewritten in vector form as follows:(8)$$P_{w}(y|o)=\frac{\exp(w\cdot F(y,o))}{Z_{w}(o)},$$
(9)$$Z_{w}(o)=\sum_{y}\exp\left(w\cdot F(y,o)\right).$$

### 3.3. Observation Probability

A motion trajectory p can be defined as a set of interconnected trajectory segments between two position points. The VINS-MONO system outputs a position point coordinate at every Δt time point. In order to reduce the number of candidate state points in Section 3.4 we used the method of the robot’s fixed length distance to extract the observation points. When the distance between the current time and the previous time is equal to a certain threshold, the visual–inertial system output is used as the observation point o(t) in the mathematical model, and the time of the sampling point is recorded simultaneously. As shown in Table 1, the record of one observation point o(t) includes the position coordinate (ξ,ψ) at time t.

Newson and Krumm proposed a map matching algorithm to find the optimal moving trajectory sequence based on the location of the anchor point and the error radius [17]. In general, map matching associates each anchor point with all the candidate road segments located within a preset error radius, as shown in Figure 5. In the method based on the conditional random field model, the position coordinate point is regarded as an observation state. Each candidate path represents a hidden state; that is, a hidden state represents a candidate state point. In this study, each candidate path represented the point closest to the observation point on the candidate path. The probability of observation depends on the distance between itself and the anchor point. It is intuitively assumed that candidate state points closer to the anchor point have higher observation probability. In the real state, there is a measurement error in the distance between the anchor point and the candidate state point, generally assuming a zero-mean Gaussian distribution. For a given observation point o(t) and candidate state point $y_{k}(t)$, the observation probability is $p(o(t)|y_{k}(t))$ and can be expressed as follows:(10)$$p(o(t)|y_{k}(t))=I(y_{k},y_{k+1})\times \frac{1}{\sigma_{1}\sqrt{2\pi }}\exp(-\frac{{\left(d(o(t),y_{k}(t))-\mu_{1}\right)}^{2}}{2{\sigma_{1}}^{2}}),$$
where $d(o(t),y_{k}(t))$ is the Euclidean distance between the observation point and the candidate status point, $\sigma_{1}$ is the standard deviation of the measured data, and $I(y_{k},y_{k+1})$ is only obtained when $y_{k}$ is connected to $y_{k+1}$. Furthermore, the state can be transferred, and the potential energy is one and blocked by the obstacle. Alternatively, if the map is far away, the state transition is not performed, and all of the transfer potentials are zero.

### 3.4. State Transition Probability

In the map, the status point may be a free space or may be occupied by a wall or an obstacle. To perform a state transition, it is necessary to know the state points adjacent to each state point, and those adjacent to the other state points [20]. In this study, the area that defined the search from the current location to the next location was called the buffer [9]. The maximum conversion distance allowed in each direction was called the buffer size. Furthermore, we set the buffer size to the distance $d_{s}$ between two state points. In the absence of any obstacles, the state transitioned between adjacent state points. In the case of obstacles, there was a conversion in the position of two buffers. Figure 6a shows the state transition when an obstacle is not present, and Figure 6b shows the state transition when an obstacle exists. The next state is shown by pink-shaded diamonds.

The transition probability refers to the transition probability between the candidate state point from time t to the candidate state point at time t+1. The robot at the candidate state point $y_{k}$ at time t is transferred to the candidate state point $y_{k+1}$ through Δt, and the shortest path passed via the Dijkstra algorithm or the Floyd algorithm is obtained, which is taken as the candidate path. The set of candidate paths between the marked observation point o(t) and the next observation point o(t+1) is $P(t)=(p_{k,k+1}(t))$, where $p_{k,k+1}(t)$ denotes a candidate path of one candidate state point $y_{k}$ of the observation point o(t) to another candidate state point $y_{k+1}$ of the observation point o(t). For two adjacent candidate state points, the transition probability can be defined as follows:(11)$$\eta (p_{k,k+1}(t))=I(y_{k},y_{k+1})\times \frac{d(o(t),o(t+1))}{l(p_{k,k+1}(t))},$$
where d(o(t),o(t+1)) is the Euclidean distance between two observation points, $l(p_{k,k+1}(t))$ is the length of the shortest path of the two candidate state points $y_{k}$ and $y_{k+1}$, and $I(y_{k},y_{k+1})$ has the same meaning as above. Considering the shortest path between candidate state points, it is possible to avoid the occurrence of detours and trajectories in the matching result.

### 3.5. Best Path Matching

The Viterbi algorithm was proposed by Viterbi in 1976 as one of the basic algorithms of the hidden Markov model [21,22]. The main problem solved by the Viterbi algorithm is to find the optimal state sequence in the sequence of state values corresponding to the sequence of observations in the case of a given model; thus, it is an optimal dynamic programming algorithm and can trace back the entire path. Suppose that t time needs to pass from S to E, there are k states $\left(y_{11},y_{12},y_{13}\ldots ,y_{kn}\right)$ at each moment; then, we only need to record the shortest path corresponding to each state. At any time, it is only necessary to consider a very limited number of the shortest paths (depending on the number of states corresponding to the moment), and there is no need to consider the previous moments upward; thus, so there is no multidimensional condition problem. As shown in Figure 7, we set t=3 and k=4 as an example to illustrate the algorithm, and the red path in the figure is the shortest path finally obtained.

In this study, it was applied to solve indoor positioning and used to find the optimal path. In the above, we used Equation (10) to get the observation probability and Equation (11) to get the state transition probability. We combined these two formulas into Equation (12) to find the most likely hidden sequence (the most likely trajectory). Finally, we used the Viterbi algorithm to solve the problem. Thus, the hidden sequence looking for the maximum probability was transformed into the path selection problem with the highest normalized probability as follows:(12)$$\begin{array}{cc} Y^{*} & =\arg\max P(y|o)=\underset{y}{\arg\max}\frac{\exp(w\cdot F(y,o))}{Z_{w}(o)}\\ & =\underset{y}{\arg\max}\exp(w\cdot F(y,o))=\underset{y}{\arg\max}(w\cdot F(y,o)) \end{array}.$$

Here, in order to considerably reduce the computational complexity and solve the problem more conveniently, there was no need to normalize the probability, and the problem could be converted into the following formula:(13)$$\max_{y}\left(\sum_{k=1}^{N}w\cdot F_{k}\left(y_{k-1},y_{k},o\right)\right),$$
where $F_{k}(y_{k-1},y_{k},o)={\left(f_{1}(y_{k-1},y_{k},o,k),f_{2}(y_{k-1},y_{k},o,k),\ldots ,f_{N}(y_{k-1},y_{k},o,k)\right)}^{T}$ is the local feature vector.

Equation (13) was solved using the Viterbi algorithm. Initialization was performed to find the non-normalized state at the most beginning position in the conditional random field as follows:(14)$$\delta_{1}(j)=w\cdot F_{1}(y_{0}=start,y_{1}=j,o),\;j=1,2,\ldots ,m.$$

Then, we recursively computed the other nodes in the conditional random field, sought the maximum conditional probability at each node, and saved the maximum path.
(15)$$\delta_{k}(j)=\max_{1\leq j\leq m}\left\{\delta_{k-1}(j)+w\cdot F_{k}(y_{k-1}=j,y_{k}=j,o)\right\},\;j=1,2,\ldots ,m.$$
(16)$$\Psi_{k}(j)=\underset{1\leq j\leq m}{\arg\max}\left\{\delta_{k-1}(j)+w\cdot F_{k}(y_{k-1}=j,y_{k}=j,o)\right\},\;j=1,2,\ldots ,m.$$

When we recursively computed the end node, the maximum conditional probability was $\max_{y}(w\cdot F(y,o))=\max_{1\leq j\leq m}\delta_{k}(j)$, the optimal path was ${Y_{k}}^{*}=\underset{1\leq j\leq m}{\arg\max}\delta_{k}(j)$, and the Viterbi algorithm obtained the optimal path as $Y^{*}={\left({Y_{1}}^{*},{Y_{2}}^{*},\ldots ,{Y_{k}}^{*}\right)}^{T}$.

## 4. Experimental Results and Analysis

### 4.1. Implementation Details

The experiments were carried out on the 10th floor of the Science Building of Beijing University of Technology with an area of 80 m × 25 m. Based on the VINS-MONO system, the preliminary trajectory was estimated using Intel Realsense d435i. In Figure 8a, the Intel Realsense D435i is a depth camera that includes a Bosch BMI055 six-axis inertial sensor, in addition to the depth camera that measures linear accelerations and angular velocities. Each IMU data packet is time-stamped using the depth sensor hardware clock to allow temporal synchronization between the gyro, accel, and depth frames. The frame rate of the RGB camera is 30 fps, and the sampling frequency of IMU is 650 Hz. Figure 8b shows the area of the experiments. We implemented all of the software in the C++/Matlab language.

Generally, to build a complete mobile robot software test platform, it needs to write a series of visual codes to display the status of the robot at all times. To realize such a complex system, it needs a lot of time and a huge amount of code work, which greatly increases the cost of building the test software system. Therefore, for the sake of time, the open-source robot operating system (ROS) was used in the test system of this paper. Finally, the algorithm was implemented in Dell G3 (the processor was an Intel Core i5 8th Gen). The experiment of this paper focused on the accuracy of location and the matching rate of the map.

Figure 9 shows the preset ideal trajectory. The black box in the lower left corner is the starting point and the end point of the trajectory, which follows the arrow in the figure. The preset ideal trajectory was constructed as the ground truth, with different unmatched trajectories, then created randomly as the input of the map matching algorithm. The map matching system was applied to refine the estimated trajectory by avoiding the crossing of obstacles. After matching the input trajectory to the map by using the CRF algorithm, we compared the matched (corrected) trajectory to the ground truth in order to measure the precision, using the errors obtained by calculating the Euclidean distance between the actual position in the ground truth and the corresponding corrected position.

### 4.2. Real Environment Experiment

In Figure 10, the moving speed of the robot was 1.5 m/s, which was very slow. The initial trajectory of the VINS-MONO system output had high robustness at the initial stage, but slight wall-crossing occurred after two turns (average cumulative error of 0.91 m). As shown in Figure 11, as a consequence of matching the trajectory with the map, the accuracy was also improved; even though the accuracy of the VINS-MONO trajectory was already good, the map matching using CRF further improved the accuracy and the cumulative error decreased. Because of the small noise and matching error of the initial positioning system, the mismatch rate was only 0.28%.

In the experiment illustrated in Figure 12, complex movements such as 180° turns and 360° turns were carried out several times, which enriched the scene transformation. In addition, some lamps in the corridor were damaged, and the light and the shade of the corridor were obviously changed, while the positioning error increased gradually with an increase in time; furthermore, the phenomenon of passing through the wall was serious (average cumulative error = 3.43 m). At this time, the moving speed of the robot was 2.5 m/s, which was relatively fast. Although the primary estimated trajectory crossed a number of walls, this was corrected successfully by using the map matching algorithm, and zero obstacles were crossed by the map matched trajectory, as shown in Figure 13. The mismatch rate was 2.94% when the initial positioning system was noisy and the error was large. As we can see, there were still some unreasonable matching points in the corner, compared with the actual trajectory of the robot; the algorithm proposed in this paper could estimate the trajectory of the robot well and meet the requirements of room-level positioning accuracy.

## 5. Conclusions

Aimed at solving the problem of the positioning accuracy of a visual–inertial system (VIS) decreasing in some indoor areas, in this study, we applied the output position of VIS to the conditional random field model by extracting the observation points and the corresponding possible state points at a fixed distance. Moreover, we made full use of the indoor structure information. In the proposed model, the Viterbi algorithm was used to find the best matching state points of the observation points in the window, finally finding the maximum probability trajectory. It fully embodied the advantages of the map matching algorithm and the probability algorithm. This algorithm was not only applicable to the VINS-MONO system; it might also be equally applicable to other visual–inertial systems. In this paper, the positioning accuracy was required to be high, but the positioning time was a little insufficient, which will be improved in the future research. For the actual scenarios, many experiments were carried out to obtain relatively good map matching results.

## Figures and Tables

**Figure 1 sensors-20-00552-f001:**
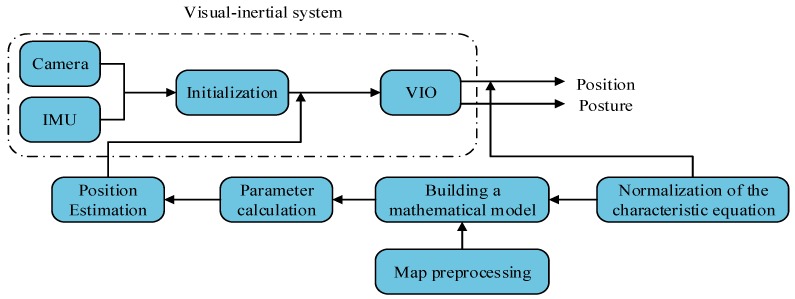
Systematic structure diagram.

**Figure 2 sensors-20-00552-f002:**
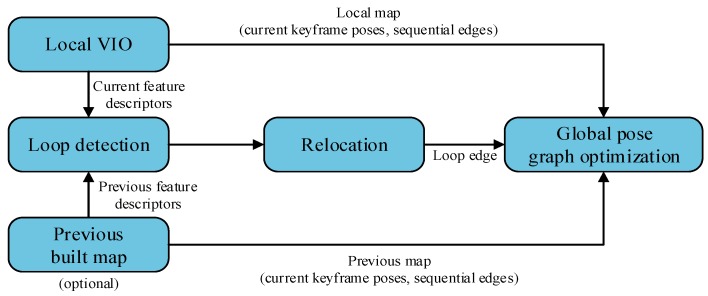
VINS-MONO system framework.

**Figure 3 sensors-20-00552-f003:**
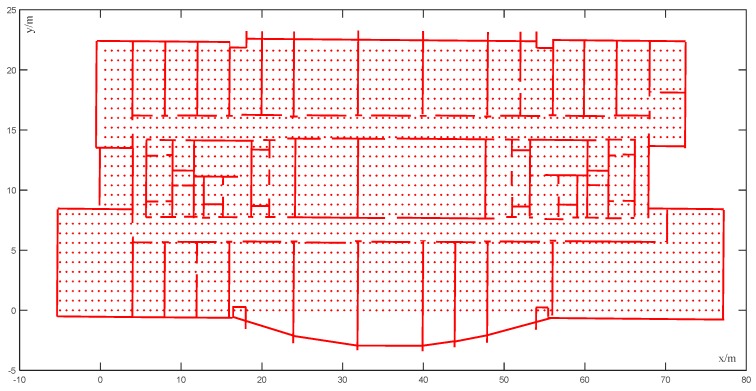
State point map.

**Figure 4 sensors-20-00552-f004:**
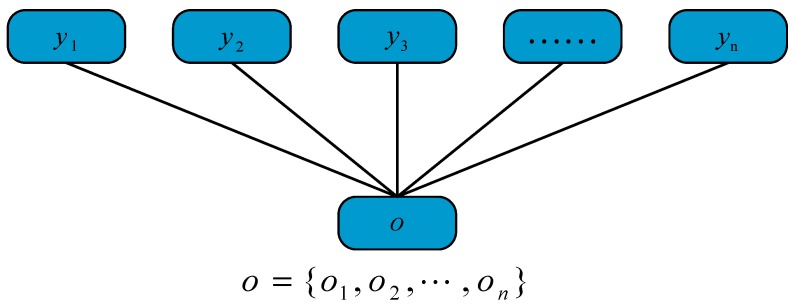
Linear-chain undirected graph model.

**Figure 5 sensors-20-00552-f005:**
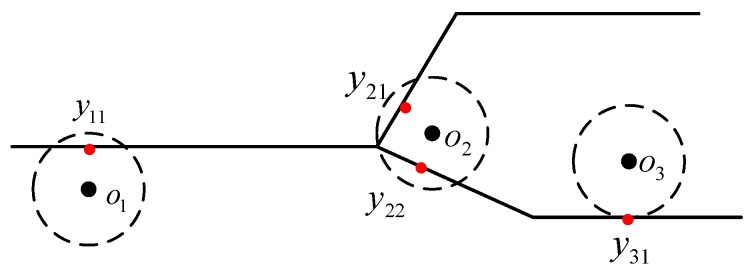
Trace diagram.

**Figure 6 sensors-20-00552-f006:**
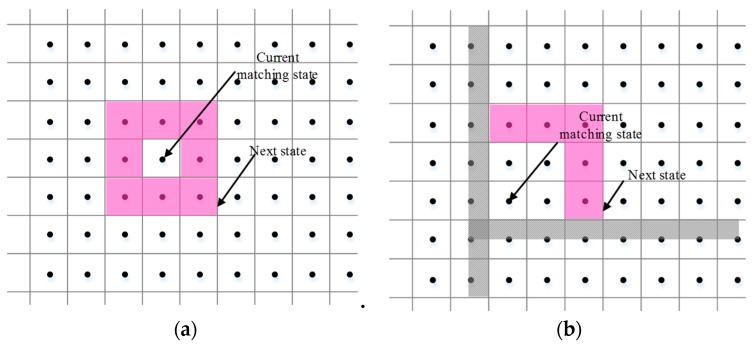
State transition diagram: (**a**) state transition when there is no obstacle; (**b**) state transition in the presence of obstacles.

**Figure 7 sensors-20-00552-f007:**
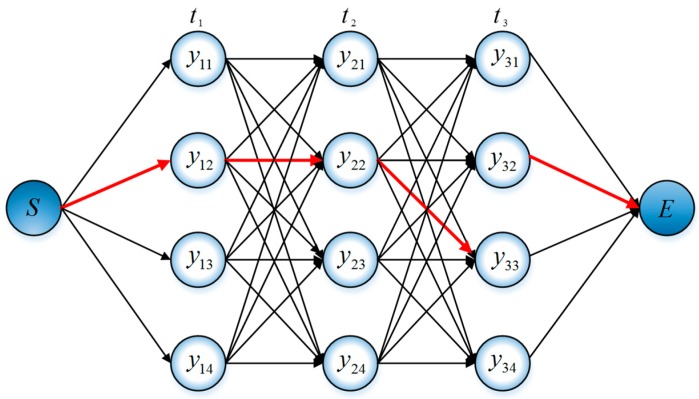
Schematic representation of the optimal path solution.

**Figure 8 sensors-20-00552-f008:**
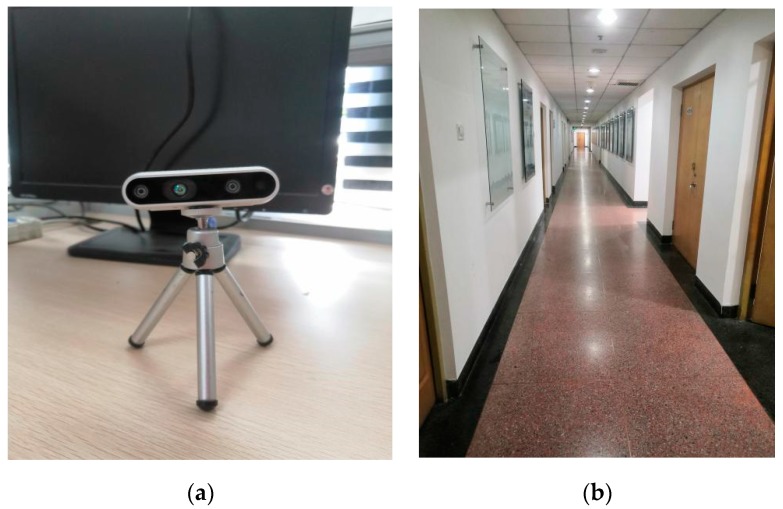
Sensors and test sites: (**a**) Intel Realsense D435i; (**b**) area of the experiments.

**Figure 9 sensors-20-00552-f009:**
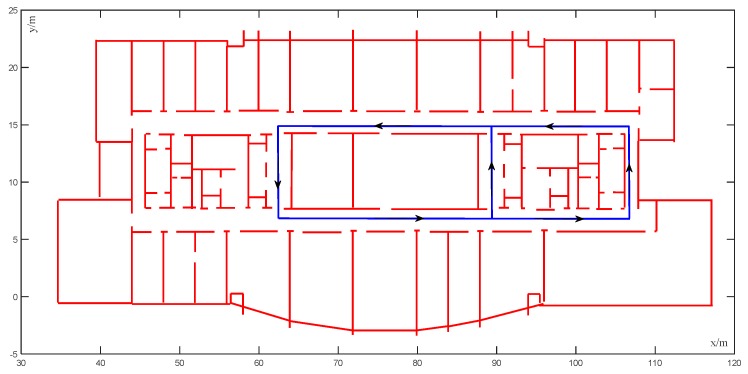
Preset ideal trajectory.

**Figure 10 sensors-20-00552-f010:**
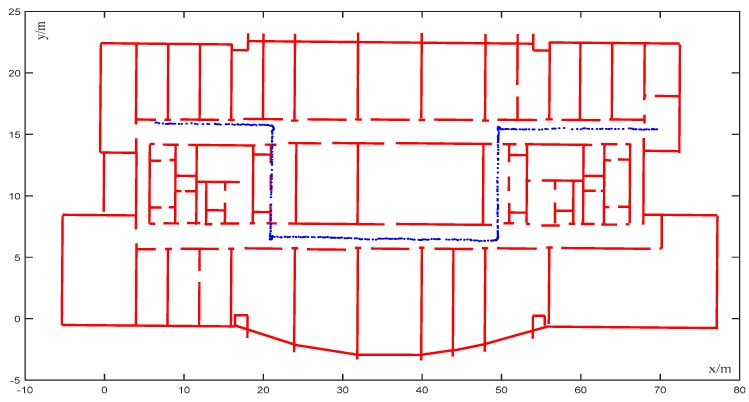
VINS trajectory with high robustness.

**Figure 11 sensors-20-00552-f011:**
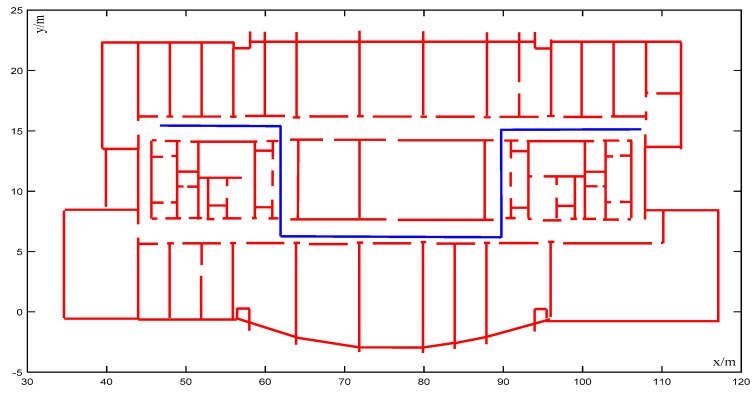
Corrected trajectory of Figure 10.

**Figure 12 sensors-20-00552-f012:**
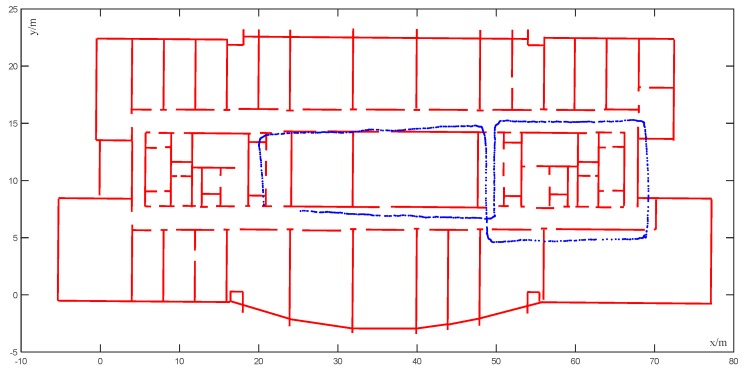
Trajectory map with large location error of VINS-MONO.

**Figure 13 sensors-20-00552-f013:**
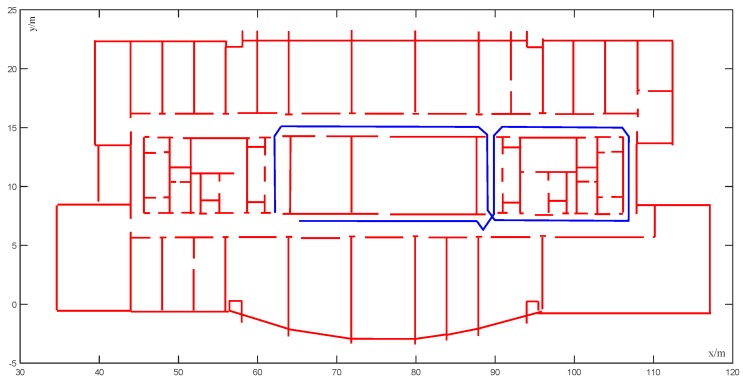
Corrected trajectory of Figure 12.

**Table 1 sensors-20-00552-t001:** VINS-MONO.

VINS-MONO Observation	(ξ,ψ)	Time
$o(t_{1})$	$\left(\xi_{1},\psi_{1}\right)$	$t_{1}$
$o(t_{2})$	$\left(\xi_{2},\psi_{2}\right)$	$t_{2}$
⋯	⋯	⋯
$o(t_{n})$	$\left(\xi_{n},\psi_{n}\right)$	$t_{n}$

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
