# Peer review of "Improving Positioning Accuracy via Map Matching Algorithm for Visual–Inertial Odometer"

_sensors, 2020, doi:10.3390/s20020552_

Round 1

Reviewer 1 Report

Dear Collegues,

the paper deals with the problem of how to improve the positioning accuracy of Visual-Inertial Odometer in closed environment. The idea proposed by the Authors is the Map Matching Algorithm in which the output of the attitude information coming from the visual-inertial odometer has been used as the input of the conditional random field model.

In my opinion the paper is very interesting because uses algorithms and methodologies, as the Viterbi Algorithm or the Markov's procecesses, typically used in completely different context.

It is true that some researchers in the past tried to follow this via, but the extensive use of the statistic made in this paper for this application is unique.

The paper is well balanced between all sections (Introduction, Mathematical Section, Experimental Section), never boring presenting all the information in clear manner and well summed up. The mathematical section is clear and complete while the experimental section gives strenght to the work.

In my opinion the paper could be already published, but in order to refine it, I suggest some little but important things:

1) Please, a part the pictures Figure 8, many figures suffer of bad resolution. Please, provide a better version for Figures 1, 2, 4, 7, 9 (here please better highlight the trajectory), 10, 11, 12, 13 (idem).

2) Check the sentences whithin the rows 88-98: at the beginning you say that the system is made of three subsections, while, at the end, you say: "fourth part is the......". Please check and improve.

3) formatting problems at rows 308-309.

4) The only thing that is not strong in this pape is the bibliography. The reference are correct but, in my opinion, a little few, therefore, let me suggest some papers that you could consider for your bibliography.

As example of possible device that have the problem to not receive information on where they are (e.g. information from satellites) even in open field, I suggest:

a) Petritoli, E., Leccese, F.
Improvement of altitude precision in indoor and urban canyon navigation for small flying vehicles
(2015) 2nd IEEE International Workshop on Metrology for Aerospace, MetroAeroSpace 2015 - Proceedings, art. no. 7180626, pp. 56-60.
DOI: 10.1109/MetroAeroSpace.2015.7180626

b) Petritoli, E., Leccese, F.
High accuracy attitude and navigation system for an autonomous underwater vehicle (AUV)
(2018) Acta IMEKO, 7 (2), pp. 3-9.
DOI: 10.21014/acta_imeko.v7i2.535

c) Petritoli, E., Leccese, F., Cagnetti, M.
High accuracy buoyancy for underwater gliders: The uncertainty in the depth control
(2019) Sensors (Switzerland), 19 (8), art. no. 1831, .
DOI: 10.3390/s19081831

As possible example of paper that use a combination of methodologies to improve the postion accuracy, I suggest:

d) Elsheikh, M., Abdelfatah, W., Nourledin, A., Iqbal, U., Korenberg, M.
Low-cost real-time PPP/INS integration for automated land vehicles
(2019) Sensors (Switzerland), 19 (22), art. no. 4896, .
DOI: 10.3390/s19224896

e) Wang, Q., Yin, J., Noureldin, A., Iqbal, U.
Research on an improved method for foot-mounted inertial/magnetometer pedestrian-positioning based on the adaptive gradient descent algorithm
(2018) Sensors (Switzerland), 18 (12), art. no. 4105, .
DOI: 10.3390/s18124105

f) Zhang, Y., Yu, F., Gao, W., Wang, Y.
An improved strapdown inertial navigation system initial alignment algorithm for unmanned vehicles
(2018) Sensors (Switzerland), 18 (10), art. no. 3297, .
DOI: 10.3390/s18103297

g) Zhang, H., Li, T., Yin, L., Liu, D., Zhou, Y., Zhang, J., Pan, F.
A novel KGP algorithm for improving INS/GPS integrated navigation positioning accuracy
(2019) Sensors (Switzerland), 19 (7), art. no. 1623, .
DOI: 10.3390/s19071623

Al the previous papers could be presented in the introduction.

Wanting to see the improvements, I'm obliged from the Journal to give a major revision even if, in my opinion, this is a little more of a minor revision.

Author Response

Thank you for your suggestions. Based on your comment and requenst, we have made extensive modification on the original manuscript. Here, we attached revised manucscript in the formats of both PDF, for your approval. A revised manuscript with the correction sections red marked was attached as the supplemental material and for easy check/editing purpose. Should you have any questions, please contact us without hesitate.

Reviewer 2 Report

This manuscript uses a visual-inertial odometer to improve the positioning accuracy of the image/ IMU integrated system, however, the title of this manuscript did not clearly show this information. The grammar and the vocabularies in this manuscript should be improved, and the moderate English changes are required. Some existing publications, (eg. Map-based indoor pedestrian navigation using an auxiliary particle filter; Motion constraints and vanishing point aided land vehicle navigation) has already used the map information, camera-derived heading information, to constrain the MEMS-based INS's solution, so using map information is not an innovation point for this manuscript. Personally, I think the main contribution is the algorithms in this manuscript, so the authors should highlight and summarize the contributions in the first section of the manuscript.   For figure 1, the authors mentioned that it is a tightly coupled system, however, from figure 1, there is no feedback on the INS  system. The letters in the figures are out of shape. line 148 and 149, the definition of an array should be improved. Figure 6 is not clear, which is the next state, one diamonds or all the pink diamonds. Most of the equations showed in this manuscript have been already derived and explained, it is not originally derived by the authors, so I highly recommend the authors use a reference instead of listing all of them in this manuscript. More tests in various kinds of environments are needed.

Author Response

(The authors gave the same response as above.)

Reviewer 3 Report

The paper is interesting and fairly easy to follow. Some remarks are given in the following.
-There is a potential typo in Fig. 2, as the central block reads "Relcalization".
-The computational complexity of the proposed solution should be analyzed, showing at least the processing time and providing information about the computer. In the current form, we have information concerning the camera frame rate and the IMU sampling frequency, but without processing time the measurement rate is actually unknown.
-Authors may compare their solution against  approaches that combine IMU units with a camera, used to detect specific graphical patterns.

Author Response

(The authors gave the same response as above.)

Round 2

Reviewer 1 Report

The paper is improved, therefore I'll suggest to accept it.